**Resource**

# ADAPTABLE: a comprehensive web platform of antimicrobial peptides tailored to the user's research

Francisco Ramos-Martín, Thibault Annaval, Sébastien Buchoux, Catherine Sarazin, Nicola D'Amelio

**Antimicrobial peptides (AMPs) are part of the innate immune response to pathogens in all of the kingdoms of life. They have received significant attention because of their extraordinary variety of activities, in particular, as candidate drugs against the threat of super-bacteria. A systematic study of the relation between the sequence and the mechanism of action is urgently needed, given the thousands of sequences already in multiple web resources. ADAPTABLE web platform (http://gec.u-picardie.fr/adaptable) introduces the concept of "property alignment" to create families of property and sequence-related peptides (SR families). This feature provides the researcher with a tool to select those AMPs meaningful to their research from among more than 40,000 nonredundant sequences. Selectable properties include the target organism and experimental activity concentration, allowing selection of peptides with multiple simultaneous actions. This is made possible by ADAPTABLE because it not only merges sequences of AMP databases but also merges their data, thereby standardizing values and handling non-proteinogenic amino acids. In this unified platform, SR families allow the creation of peptide scaffolds based on common traits in peptides with similar activity, independently of their source.**

## Introduction

Antimicrobial peptides (AMPs) are a class of molecules that have attracted significant attention for their antibacterial properties (Wang et al, 2015). As part of organisms' innate immunity, AMPs have been found virtually in all life kingdoms, including marine and terrestrial animals, bacteria, and plants (Zasloff, 2002; Goyal & Mattoo, 2016; Wang et al, 2017). However, particular attention has been devoted to those produced by animal venoms and frog and toad skin secretions (Xu & Lai, 2015). They display activity against a wide range of targets: bacteria, viruses, fungi, parasites, insects, and cancer cells (Tamamura et al, 1998; Albiol Matanic & Castilla, 2004; Hoskin & Ayyalusamy, 2008; Joo et al, 2012; Lacerda et al, 2014; Yang et al, 2014; Field et al, 2015). They also modulate inflammation processes and cell–cell communications (Oyinloye et al, 2015). Thousands of AMPs are already known (Aguilera-Mendoza et al, 2015), but the number is destined to grow as hundreds of new peptides are discovered or synthetically produced each year (Wang et al, 2015). It is apparent that fast throughput methods are needed to enable their mechanism of action to be determined.

### A novel tool in the era of antimicrobial resistance

ADAPTABLE (Antimicrobial PeptiDe scAffold by Property alignmenT. A weB platform for cLustering and dEsign; http://gec.u-picardie.fr/adaptable) was created to provide a tool for all scientists in the field of AMPs who are interested in developing new drugs against a well-defined target. For example, according to the World Health Organization, there are 12 microorganisms considered the greatest danger towards human health (Vogel, 2017) because of their resistance to known antibiotics. For each microorganism, ADAPTABLE is able to generate SR families containing peptides known to be active against it and classify them, thus highlighting essential traits. Alternatively, each of these peptides can be used as a bait to generate their own SR families by scanning the ADAPTABLE database featuring more than 40,000 entries. Although sequence alignment has been widely used among AMPs with strictly related biological source, we believe that the comparison of evolutionarily distant AMPs with similar activity can highlight the features that are key to their mechanism of action. This unbiased search can be used to spot existing, but untested, sequence-related peptides of similar structure and mechanism of action that also have the properties of more-promising drug candidates (e.g., less hemolytic).

### A different classification strategy based on property alignment

Expression of many similar AMPs by the same cell type, organism, or genus has resulted in AMPs to be classified in families based on sequence alignment of peptides with closely related biological origin (Zouhir et al, 2010; Tassanakajon et al, 2015; Xu & Lai, 2015). However, common features can be found in families of all origins, such as (i) the recurrent presence of positively charged residues, thought to draw the peptide towards the negatively charged membranes of bacteria and cancer cells, and (ii) a significant fraction of

Génie Enzymatique et Cellulaire, Unité Mixte de Recherche 7025, Centre National de la Recherche Scientifique, Université de Picardie Jules Verne, Amiens, France

Correspondence: nicola.damelio@u-picardie.fr; francisco.ramos@u-picardie.fr

hydrophobic residues facilitating the interaction with the lipid bilayer. This suggests that the clustering of sequences with specific activities (what we call "property alignment") independent of the evolutionary distance can highlight those key features underpinning mechanisms of action.

Several methodologies and software already exist for the classification and prediction of AMP activities (Brahmachary et al, 2004; Fjell et al, 2007; Wang et al, 2011; Ng et al, 2015; Bhadra et al, 2018). Most provide tools for sequence similarity searches, and only few have algorithms for predicting the activity of any given peptide sequence (e.g., APD [Wang, 2014], CAMPR3 [Waghu et al, 2016], AVPpred [Thakur et al, 2012], YADAMP [Piotto et al, 2012], and ToxinPred [Gupta et al, 2013]). However, none of these combine a systematic global classification with "adaptability" (hence, the name ADAPTABLE) to the focus of the user's research.

### A different approach tailored to researchers' aims

One important difference of ADAPTABLE with respect to other alignment tools is that it provides the facility to study peptides active against a specific pathogen, even when a researcher does not already have a specific peptide of interest. ADAPTABLE generates many different SR families with activity against a specific pathogen, optionally including one or more peptides provided by the user. The members of different SR families are significantly different in sequence and possibly represent different mechanisms of action. Each SR family is represented by the peptide used as bait for its creation, also called "father." The researcher can then study each father to get insights into the molecular causes of the activity and/ or concentrate on the SR family that his own peptide has been assigned to.

### A comprehensive and standardized database

Whereas the power of the algorithms relies on large number of entries, the concept of property alignment requires standardization. In other words, the development of ADAPTABLE web platform has implied the creation of a comprehensive and standardized database that addresses a further fundamental problem in the study of AMPs: information scattering across multiple web resources. Existing databases have previously been merged (Aguilera-Mendoza et al, 2015), but previous standardization only collected and merged sequence information. In contrast, ADAPTABLE merges information for more than 60 properties from 25 databases specialized in both medical and agricultural fields where available and creates a unified, nonredundant entry. This way, each sequence is associated to all available data on its structure (predicted or experimental), activity (e.g., anticancer, antibacterial, antibiofilm, antiviral, antifungal, or antiparasitic), targets (e.g., lung cancer, HeLa cells, HIV virus, cell membrane target, *Klebsiella pneumoniae*, or *Plasmodium falciparum*), and other information (e.g., experimental validation, taxonomy, bibliography, or DSSP—"define secondary structure of proteins" [Kabsch & Sander, 1983] data).

Merging of data from different sources requires standardization. This task can be challenging because each existing database follows its own format and definitions. For instance, activities are reported in many different units or measured by different tests. Our

algorithm converts all activities to micromolar (μM) concentration by calculating the molecular weight of the peptide, even for non-proteinogenic amino acids. Even if different activity values are not always directly comparable because of different experimental conditions or activity tests, an upper threshold value allows the filtering out of less active peptides, while the "activity test" toggle allows restriction to a single test.

To our knowledge, ADAPTABLE is the only tool able to standardize activities and non-proteinogenic amino acids, including modified and non-natural amino acids. Non-proteinogenic amino acids lack a standardized one-letter code, resulting in confusion when comparing sequences across different databases. This ambiguity often leads to redundant entries referring to the same sequence being nonuniformly annotated. Furthermore, some non-proteinogenic amino acids are named by different synonyms rather than a unified nomenclature that, if it existed, would allow unambiguous identification. ADAPTABLE addresses both issues by interpreting the different nomenclatures and providing a single name based on the PubChem database (Kim et al, 2019) and a single one-letter symbol. For non-proteinogenic amino acids, the closest proteinogenic homologue is also identified, where possible.

ADAPTABLE takes the standardization process one step forward, thanks to its inclusion of data from a specialized microbiology database ("The Microbe directory" [Shaaban et al, 2018]). This allows the inclusion of potentially missing information such as the full names of organisms, their nature (i.e., Gram positive or negative bacteria, fungi, or virus, among others), and some of their properties (i.e., ability to form biofilms). The same approach is followed to complement and store structural information, either predicted or experimental, leveraging ADAPTABLE's integration with the Protein Data Bank (Berman et al, 2000) and the usage of PSSpred (Yan et al, 2013) and I-TASSER (Roy et al, 2010; Yang et al, 2015; Yang & Zhang, 2015).

# Results and Discussion

### Family generator tool

#### Defining the subset of peptides

ADAPTABLE offers the possibility to generate SR families using subsets of peptides with user-defined characteristics ("Family Generator" page shown in Fig 1; for details, see the case example n.1, "Designing new peptides active towards a specific organism and highlighting motifs", in Supplemental Data 1, "ADAPTABLE Tutorial"). This tool requires a "calculation label" and a "username" that allow the user to privately analyze ("Family Analyzer" page) or download the results ("Download Results" page) at the end of the calculation. Email is used to notify the user about the start and the end of the run, together with the "Calculation label".

The subset of peptides can be defined by their name, sequence pattern, and target organism. The expandable "Advanced" and "Peptide properties" sections provide further tuning of the required peptide properties. By selecting from among the more than 60 parameters, the user can choose the target organism, activities, or chemical or physical properties, and also stipulate other parameters (source, taxonomy, posttranscriptional modifications, N-terminal or C-terminal

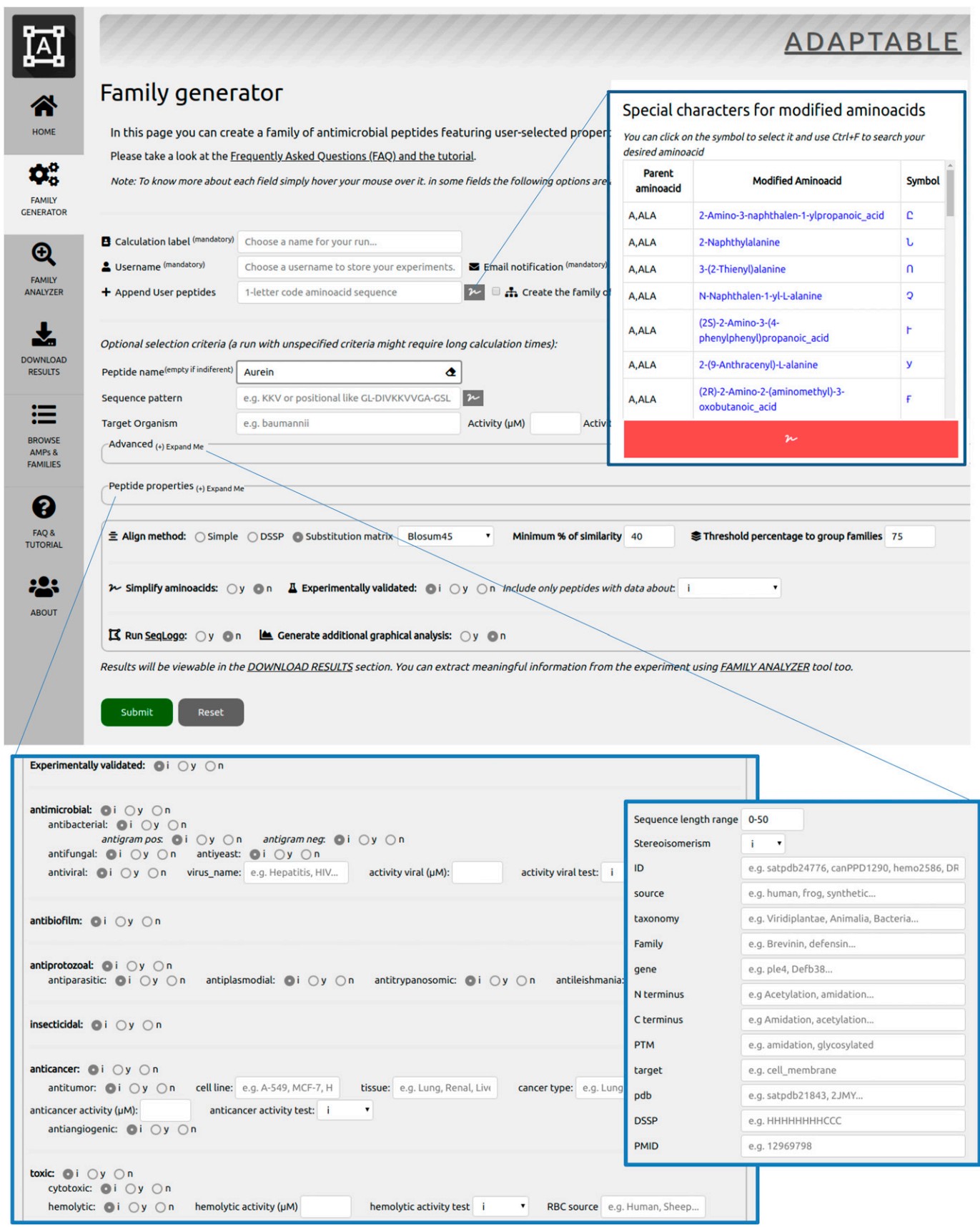

modifications, solubility, etc.). For example, switching on the "Experimental structure" option will filter out all peptides whose structure has not been experimentally obtained.

It is also possible to restrict the selection to those peptides tagged "experimentally validated" by their original source database ("Experimentally validated" toggle). Further restriction is possible by selecting only peptides for which (i) the target organism is described or (ii) the target organism is described and a specific value of activity has been measured.

The user can optionally include their own sequences in the calculation ("Append User peptides"). A remarkable feature of ADAPTABLE is its ability to handle non-proteinogenic amino acids in a standardized fashion. Thanks to a character picker (insert at the top of Fig 1), the user can insert non-proteinogenic amino acids in the sequence by clicking their symbol. The "Simplify amino acids" option transforms non-proteinogenic amino acids to their most similar proteinogenic counterpart, if possible.

Rather than selecting peptides based on their properties, the user can choose to create an SR family using a specific reference sequence ("Create the family of a specific peptide"). This feature can be very useful for classifying new peptides of unknown activity. If very similar sequences are found in the database, this option can even be used to help determine their biological properties. For example, if the peptide introduced by the user has generated an SR family that is 80% antibacterial and 70% anticancer, it can be hypothesized that this peptide potentially has both activities, providing an interesting hypothesis to validate experimentally.

### Choosing the alignment method

ADAPTABLE allows three kinds of alignment methods: "Substitution matrix," "Simple," and "DSSP." The first (default) method uses mutation substitution matrices. The user can choose from among multiple point accepted mutation and BLOSUM matrices or the simpler unitary scoring matrix, with the possibility to edit both the minimum percentage of similarity for the generation of the SR families and the minimum percentage of common peptides to group similar SR families. The "simple" option was introduced to highlight very general properties shared by evolutionary distant peptides (i.e., the presence of amphipathic helices). The aim of the simple mode is to drastically reduce the number of amino acid types, and it will convert any amino acid to one of the nine classes: hydrophobic residues (A, V, I, L, and M) represented by A, negative residues (D and E) by D, positive residues (K and R) by K, aromatic residues (W, Y, H, and F) by F, polar residues (S, T, N, and Q) by S, and modified amino acids by Ⓜ. Gly, Pro, and Cys are treated individually.

Finally, the "DSSP" option aligns sequences on the basis of the secondary structure to highlight the role of well-defined three-dimensional arrangements. Seven conformations are present in the DSSP notation (Kabsch & Sander, 1983): $\alpha$-helix (H), 3–10 helix (G), $\pi$ helix (I), $\beta$-bridge (B), $\beta$-strand (E), turn (T), and bend (S).

### Additional graphical analysis

ADAPTABLE optionally computes a series of properties during the generation of SR families and creates multiple visualizations of the results. Besides peptide length distribution, ADAPTABLE computes for each SR family the average presence of proteinogenic amino acids, their average number per peptide, and their percent occurrence at each position. The average presence may highlight the importance of specific amino acid types. For example, positively charged residues are commonly present in AMPs, and they are thought to drive peptides towards negatively charged bacterial membranes; the average number per peptide (e.g., two cysteines) might suggest the presence of specific interactions (e.g., disulphide bonds). The rate of occurrence of each amino acid in each position introduces the spatial information, which is also essential for the design of new active peptides. For example, it has been shown that the distribution of hydrophobic amino acids along the sequence is highly asymmetric in selective AMPs, whereas it tends to be rather constant in hemolytic peptides (which tend to insert deeper in the membrane of the host) (Juretić et al, 2011). A more detailed description about this concept is present in ADAPTABLE Tutorial (Supplemental Data 1).

### Family analyzer

The "Family Analyzer" page provides an overview of the properties of the SR families originated by the "Family Generator." This tool provides both a summary of the SR family properties (Fig 2, top) and a full output (Fig 2, bottom). The summary page displays statistical analyses of the properties of the SR family and lists each member's sequence with links to the ADAPTABLE and source database entries. The full output contains an interface, allowing rapid and effective navigation of a large amount of data: for example, it is possible to visualize sequence alignment with color codes for residue conservation (frequency), polarity, amino acid types, or secondary structure. Alternatively, one may choose to visualize only selected information such as the source and the gene of origin of each member of the SR family. A more detailed description of this tool is given in the Tutorial, Supplemental Data 1.

### Downloading results

Results can be downloaded by providing the username and calculation label. Data are available for 6 mo. Once downloaded, the HTML output can be read in a browser.

### ADAPTABLE browsing tool

The ADAPTABLE browsing tool ("Browse AMPs & Families") is composed of three subsections:

---

**Figure 1. Screenshot of the ADAPTABLE "Family Generator" page.**
A calculation label and username are provided to allow private access to the results. The user can optionally append new sequences to the calculation. The "Character picker" (insert above) allows the insertion of special characters used for non-proteinogenic amino acids into the sequence fields. The picker offers a direct link to the PubChem (Kim et al, 2019) database for each entry. Expanding the "Advanced" and "Peptide properties" sections (see inserts in the bottom) offers the possibility to choose from among over more than 60 parameters. In the last part of the page, the user can select the desired alignment method, the minimum similarity threshold for the creation of SR families, and the minimum percent overlap among SR families for their clustering. Additional graphical analysis is optionally offered.

# Summaries

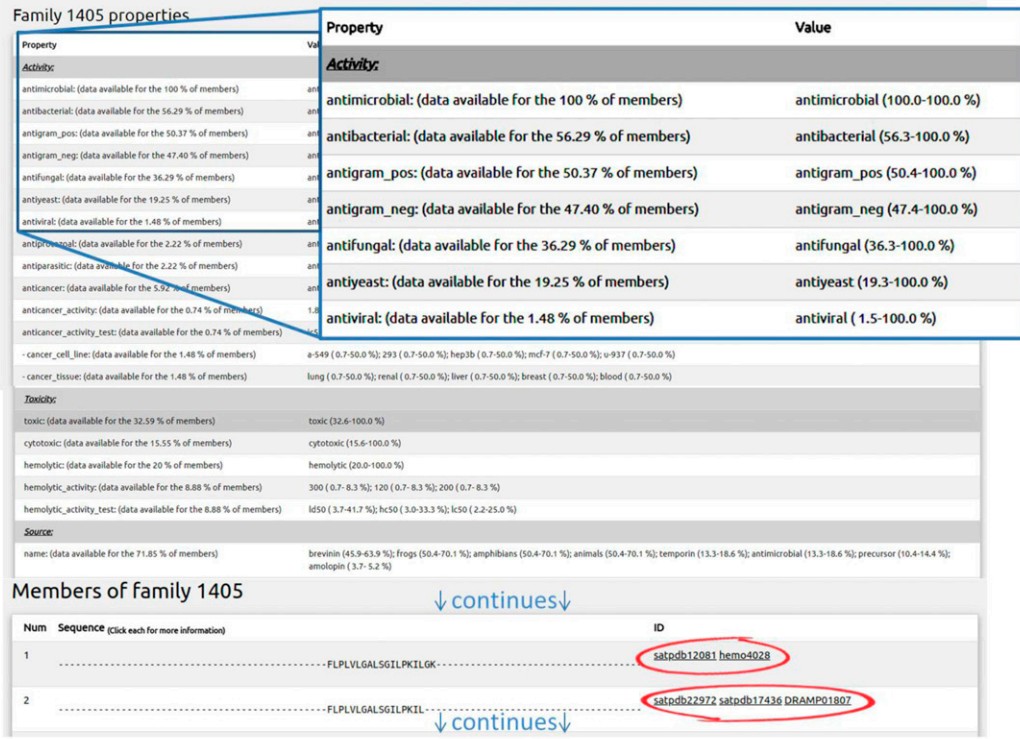

# Full output

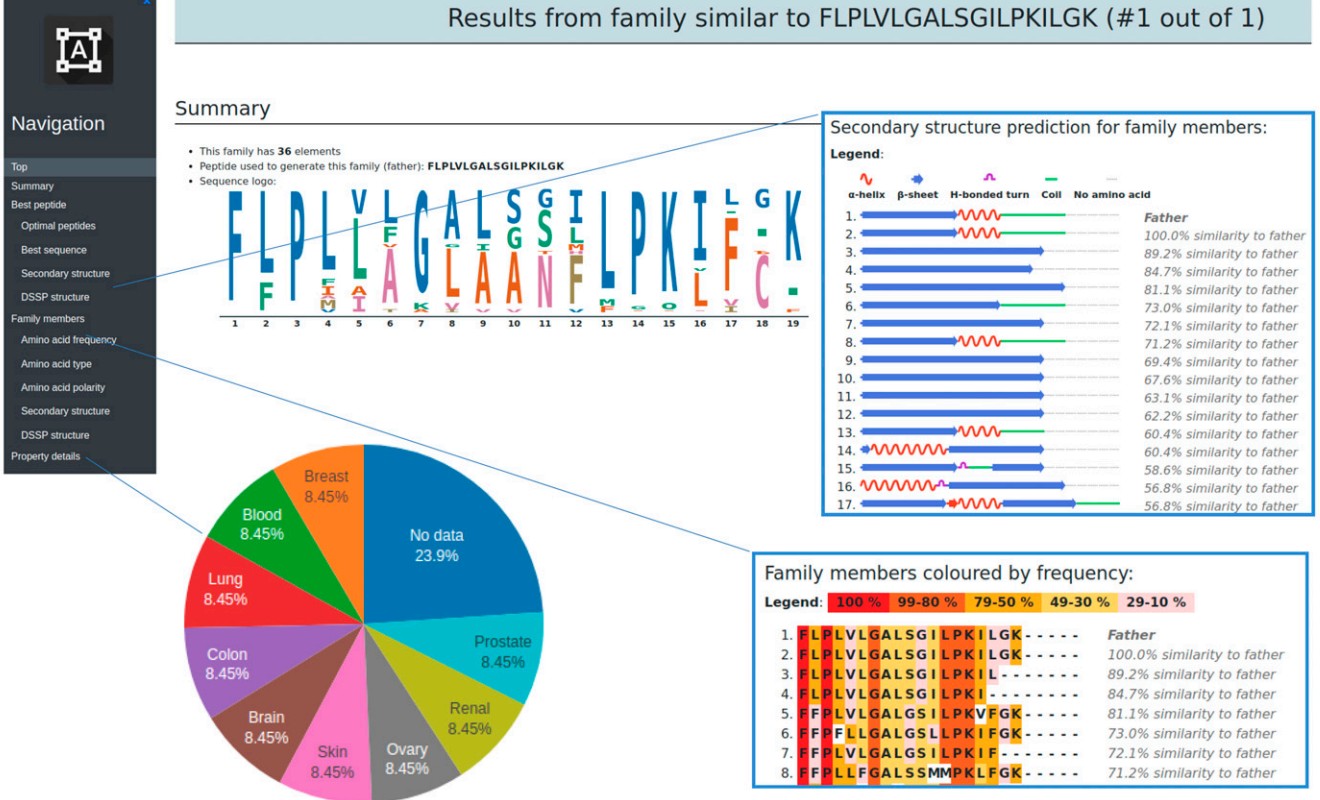

The first one, "Browse AMPs Database," offers the possibility to view single entries by typing part of a sequence or name; the ADAPTABLE entry consists of a table summarizing all available data on the selected peptide and providing links to the related SR families obtained in the "all_families" built-in experiment described below (see Screenshot 2 in the Tutorial, Supplemental Data 1). In particular, the first field is a link to the SR family generated by the peptide, whereas the second field contains links to all SR families where the peptide can be found. The field "External database ID" can be used to visit the peptide entries in other databases. When available, biochemical parameters (provided by ExPASy ProtParam tool [Gasteiger et al, 2005]) and the relevant literature are accessible by links. Direct access to the Protein Data Bank is provided for the experimental tridimensional structure, whereas integration with I-TASSER (Roy et al, 2010; Yang et al, 2015; Yang & Zhang, 2015) allows structural prediction even when the experimental one is not available.

The second subsection, "Family overview," allows finding information regarding the built-in "all_families" experiment, created by sequence alignment of all the peptides in the database. It constitutes an unrestrained version of the general procedure for the creation of any ADAPTABLE SR family, where no restrictions are applied by the user in terms of peptide properties. As a consequence, each peptide acts as a father for the formation of a family.

The third subsection allows generation of a FASTA file with the sequences of peptides featuring user-defined properties.

### Future Directions

With ADAPTABLE, we want to provide the researcher with a tool to study the mechanism of action of AMPs. Its completeness in terms of database sources permits its applications in various research fields, spanning from medicine to agriculture, food preservation, and antiseptic materials.

We have shown how the generation of SR families is able to group peptides based on user-defined characteristics. ADAPTABLE also has built-in functions to localize conserved residues, thus highlighting motifs (well-definite arrangements of amino acids likely to be responsible for the different activities). The motifs are created by taking into account intrasequence correlations to avoid nonfunctional chimeras. This function is designed to provide optimal scaffolds for drug design. Recent studies suggest that this approach could be very useful for scientists working in this area (Schmitt et al, 2016; Almaaytah et al, 2017). We believe that the architecture of ADAPTABLE, besides offering easy access to all available information on a specific peptide described in many different databases, can also be used by the scientific community to (i) design new peptides using motifs responsible for the specificity towards a specific organism, (ii) predict several properties of a generic sequence, (iii) discover experimentally untested activities

for a given peptide by retrieving information on similar sequences in its SR family, and (iv) generate optimal scaffold for drug design, thanks to the generation of the SR family representing sequence.

ADAPTABLE has been designed as a self-updating platform, thanks to its automated tools that aggregate data from upstream sources. This is a fundamental characteristic because we expect that the number of peptides will continue to increase in the following years. ADAPTABLE will, therefore, continue to incorporate more AMPs and databases. To this end, we have established a simple text input that allows external contributors to translate their data into the ADAPTABLE format, whose structure is shown at http://gec.u-picardie.fr/adaptable/faq.html#newdb. We subscribe to a commitment-to-updates that involves being responsible for maintaining the website and updating the resources regularly.

### Additional Files

The ADAPTABLE tutorial file (http://gec.u-picardie.fr/adaptable/ADAPTABLE_tutorial.pdf#view=FitH) contains specific case examples guiding the user step-by-step through the calculation and interpretation of results.

# Materials and Methods

### ADAPTABLE AMPs database

ADAPTABLE incorporates automated tools to periodically download, process, and merge data to keep synched with data sources: ADAM (Lee et al, 2015), ANTISTAPHYBASE (Zouhir et al, 2017), APD (Wang & Wang, 2004; Wang et al, 2009; Wang et al, 2016), AVPdb (Qureshi et al, 2014), BaAMPs (Di Luca et al, 2015), BACTIBASE (Hammami et al, 2007, 2010), CAMPR3 (Waghu et al, 2016), CancerPPD (Tyagi et al, 2015), ConoServer (Kaas et al, 2008, 2012), CPPsite (Gautam et al, 2012; Agrawal et al, 2016), DADP (Novković et al, 2012), DBAASP (Gogoladze et al, 2014; Pirtskhalava et al, 2016), Defensins (Seebah et al, 2007), DRAMP (Fan et al, 2016; Kang et al, 2019), Hemolytik (Gautam et al, 2014), HIPdb (Qureshi et al, 2013), InverPep (Gómez et al, 2017), LAMP (Zhao et al, 2013), MilkAMP (Théolier et al, 2014), ParaPep (Mehta et al, 2014), Peptaibol (Whitmore & Wallace, 2004), PhytAMP (Hammami et al, 2009), SATPdb (Singh et al, 2016), UniProt (The UniProt Consortium, 2018), YADAMP (Piotto et al, 2012), PubChem (Kim et al, 2019), The Microbe Directory (Shaaban et al, 2018), and the Protein Data Bank (Berman et al, 2000).

Data are merged in a single database of more than 40,000 unambiguous sequence entries. In the future, even more databases and resources will be implemented to enrich the information available for each sequence. Third-party contributors can develop their own tools to generate a final output adhering to our standardized simple text format (see "FAQ and Tutorial" section of the Supplemental Data 1).

---

**Figure 2. Screenshots of "Summary" (top) and "Full output" (bottom) for each SR family.**
The Summary (top) provides the minimal percentage of members displaying each property and, for each member, the links to their ADAPTABLE single entry, and to external databases. The full output (bottom) shows the properties in detail for each member and the peptide best representing each family. The SR family members can be visualized sorted by similarity to the father and colored by amino acid frequency, residue type, polarity, predicted secondary structure, or DSSP prediction (see inserts). Multiple pie charts give a visual representation of each distinct property.

### SR family generator algorithm

After the user has selected interesting peptides based on properties and activity, ADAPTABLE creates SR families by comparing each sequence with all others by pairwise alignment (Fig 3). By default, the sequence similarity score is computed by applying mutation data matrices (Oren & Shai, 1998; Huang et al, 2010). The choice of BLOSUM45 ("BLOcks of Amino Acid SUbstitution Matrix") as a default value was mandated by the fact that the database contains AMPs from very different sources, which are, thus, evolutionarily distant. In addition, other BLOSUM, "point accepted mutation" or unitary (Dayhoff et al, 1983) (1 or 0 score based on identity) matrices can optionally be chosen. Alignment in "simple" or "DSSP" modes (see the Results and Discussion section) uses the standard unitary scoring matrix.

No gaps or insertions are allowed for several reasons. Most AMPs display helical structures, allowing its biological function (Bechinger, 1996; Oren & Shai, 1998; Vogt & Bechinger, 1999; Huang et al, 2010). For example, alternation of polar and non-polar amino acids in the primary sequence allows the formation of amphipathic helices capable of forming channels in bacterial membranes (Aisenbrey et al, 2019). In helical structures, sequence insertion and deletion result in severe alteration of the relative orientation of all subsequent amino acids and are, therefore, unlikely. Furthermore, allowing insertion and deletion in short sequences would introduce too much variability, ultimately masking the detection of short local motifs that are the focus of ADAPTABLE (Myers, 1991; Giegerich & Wheeler, 1996).

Once a full comparison has been accomplished, ADAPTABLE has generated one SR family for each peptide. These SR families are sorted by their number of elements, with the first SR family being the largest in size. Some SR families are "relatives" in the sense that they share subsets of peptides. The SR families can be gathered in groups by defining the percentage of peptides in common (parameter

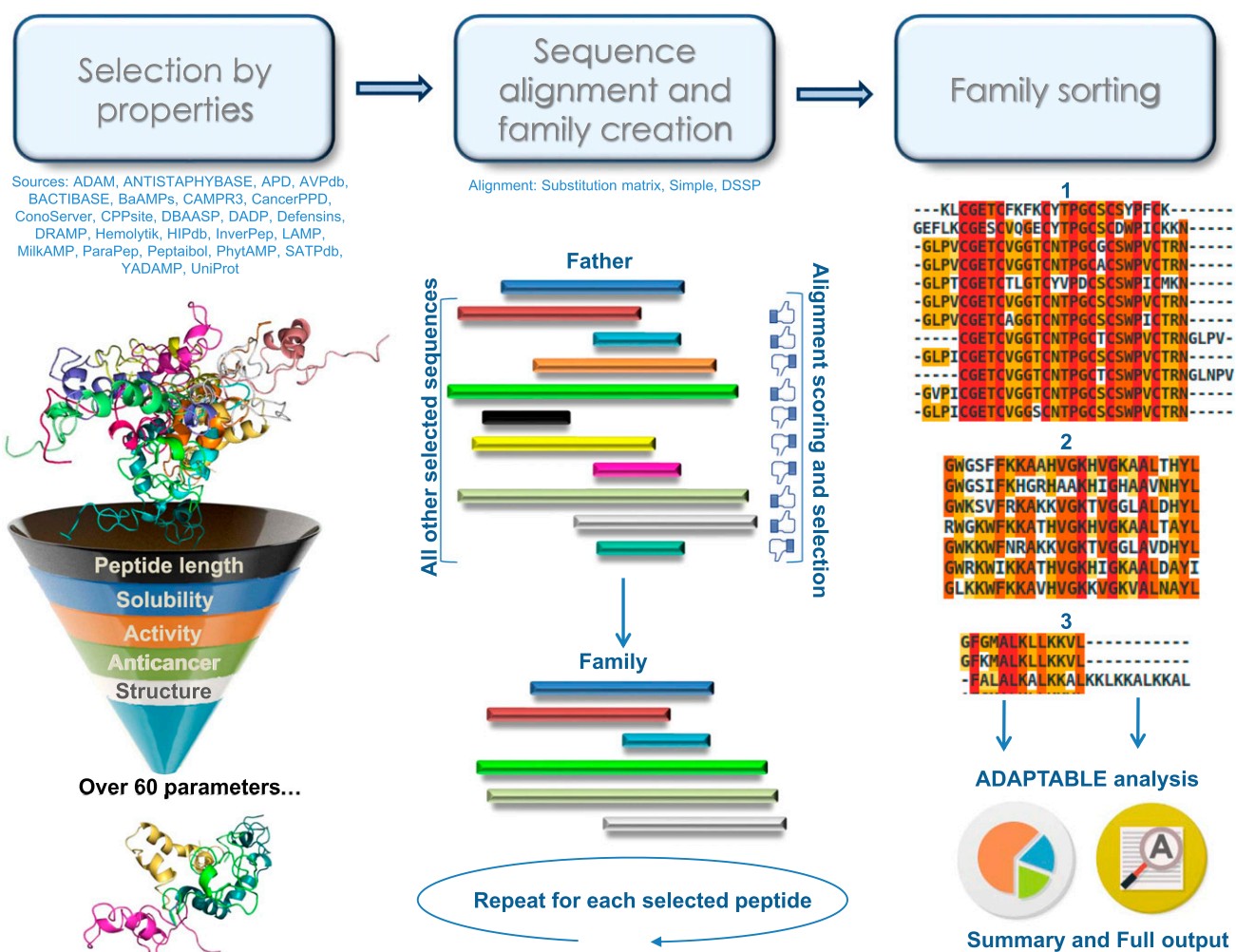

**Figure 3. ADAPTABLE SR family generation procedure.**
In the first step (left), the user selects the desired properties of AMPs from among multiple optional parameters. Starting from the more than 40,000 participating peptides, each selection acts like a funnel and restricts the number that goes forward to the next stage; the creation of SR families. Each of the selected peptides is tested as an SR family father (central image) and compared with all the other selected peptides in all possible relative alignments. Peptides displaying sequence similarity (thumbs up in the figure) are retained within the father's SR family, and the process is repeated for each subsequent peptide in turn as a potential father. The alignment can be performed by three methods: simple, DSSP, and substitution matrix (see main text). In the last step (right), SR families are sorted according to their size (and colored according to residue conservation) and processed to give the "summary" and "full output," as described in "Results and Discussion."

"Threshold percentage to group families" in the "FAMILY GENER-ATOR" webpage).

In particular, the largest SR families in a group are originated by fathers that are compatible with the largest number of entries and, therefore, contain only the essential traits. ADAPTABLE sorts SR families by size so that a larger family index might correspond to increasing activity specificities.

For example, consider SR families 3 and 8: they are in the same group and SR family 3 is originated by a father containing the motif XZ-YYY (where X, Y, and Z are amino acid types) that acts upon biomembranes via a carpet mechanism. The elements of a similar, but smaller, SR family (SR family 8) generated by a father containing the motif XZY-YYY might act via the same mechanism but with preference for Gram-positive bacterial membranes. Such a finding would prompt the researcher to study the two fathers to better understand the molecular-level causes of the increased selectivity.

### Generation of the SR family–representing sequence

Specific arrangement of amino acids (motifs) is often correlated with the local structure (Richardson & Richardson, 1988), the interaction with other partners, or the stability of the peptide (Ashenberg et al, 2013). Although the father constitutes a good representative of each SR family, it might contain parts which are not essential traits of the ensemble. For this reason, ADAPTABLE generates a representative peptide for each SR family of sequences to highlight only the essential motifs, those from which the activity could originate. To preserve the information contained in the residue conservation across the SR family, without losing the information on intramolecular interactions within each sequence, ADAPTABLE generates one single template peptide representing the family, starting from each most abundant residue "$aa$" at position $p$. Each sequence is built by using the most frequent amino acid found at distance $p'$ within the peptides having "$aa$" at position $p$ (if the frequency is below 10%, the position $p'$ is considered variable and replaced by a dash). Finally, the most-representative peptide is chosen: the one displaying the maximum value of the sum of frequencies at each position $p$.

### Motifs (position-dependent and position-independent)

The method for generating the representative sequence (that explained above) is a good tool for highlighting motifs because it reflects the mutual distance of amino acid types within a peptide. The representative sequence contains position-dependent motifs because they are drawn on well-defined positions (e.g., –X-YX————ZZ-X with X, Y, and Z generic amino acid types, and X-YX and ZZ-X as two independent motifs).

Information of even greater significance derives from the position independence of the method; that is, the occurrence probability of amino acid $j$ at distance $d$ from amino acid $i$ is calculated independently using the position of $i$ in the sequence. The motifs are then called position-independent, and ADAPTABLE reports their analysis in the graphical output ("Generate additional graphical analysis" option set to "y" in the "Family Generator" page).

### Property calculations

ADAPTABLE features a simple algorithm to predict solubility in water based on the number of hydrophobic (I, L, M, F, V, W, and Y) relative to charged (K, R, H, D, and E) residues. Its prediction outputs are soluble (less than 5 amino acids or <50% hydrophobic and >25% charged), poorly soluble (<50% hydrophobic and ≤25% charged), almost insoluble (50–75% hydrophobic), and insoluble (hydrophobic ≥75%).

It also performs a prediction of the secondary structure based on the Chou–Fasman method (Chou & Fasman, 1974, 1978, 1979).

### Additional graphical analysis

Besides the analysis of motifs and of the SR family–representative peptide, other types of graphical outputs can be created by the "Family Generator" tool ("Generate additional graphical analysis" option set to "y") that reports on the amino acid composition and position in the sequence.

The sequence logo generation based on WebLogo (Schneider & Stephens, 1990; Crooks, 2004) can be performed when the "Run SeqLogo" option is set to "y" in the "Family Generator" page. Examples are given in the Tutorial (Supplemental Data 1).

### Implementation

ADAPTABLE is developed using mainly GAWK (version 4.1 or newer at the time of the writing) (Robbins, 2003) and Bash (4.3 or newer) (Ramey & Fox, 2015). Graphics generation relies on Matplotlib (2.2 or newer) (Hunter, 2007) and Python (3.5 or newer) (Van Rossum & Drake, 2011). Parallelization is achieved using GNU parallel (Tange, 2011) and Python Joblib. The web interface relies on Javascript and CSS3 and adheres to HTML5 web standard, running Apache 2.4 server on a Linux system.

### Availability of Supporting Source Code and Requirements

- Project name: ADAPTABLE.
- Project home page: http://gec.u-picardie.fr/adaptable/.
  - https://gitlab.com/pachoramos/ADAPTABLE.
- Operating system(s): Platform independent (client runs in the web browser).
- Programming language: AWK, Bash, Python, HTML5, and Javascript.
- Other requirements: Web browser adhering to HTML5 standards (i.e., Chrome, Firefox, Safari, and Edge).
- License: EUPL-1.2.

# Supplementary Information

# Acknowledgements

We would like to thank Professor Manuel Dauchez, University of Reims Champagne-Ardenne, Matrice Extra-cellulaire et Dynamique Cellulaire (MEDyC),

Unité mixte de recherche 7369, Centre national de la recherche scientifique for useful discussion, and the Matrics platform at the University "Picardie Jules Verne" for providing computing resources. We would also like to thank Graham Bentham for English language editing. Funding: This work was partly supported by the Université de Picardie Jules Verne, S2R 2018 - Action 1: "Incitations au dépôt de projets de recherche"; Francisco Ramos-Martín's PhD scholarship was co-funded by Conseil régional des Hauts-de-France and by *European Fund for Economic and Regional Development* (FEDER). The authors are grateful to the Université de Picardie Jules Verne for its financial support for publication through its S2R action.

## Author Contributions

F Ramos-Martín: conceptualization, data curation, software, validation, visualization, methodology, and writing—original draft.
T Annaval: data curation, software, methodology, and writing—review and editing.
S Buchoux: software, visualization, methodology, and writing—review and editing.
C Sarazin: conceptualization and writing—review and editing.
N D'Amelio: conceptualization, data curation, software, formal analysis, supervision, funding acquisition, validation, project administration, and writing—original draft, review, and editing.

## Conflict of Interest Statement

The authors declare that they have no conflict of interest.

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
