## [Reviewer comments · Life Science Alliance]

ADAPTABLE: a comprehensive web platform of antimicrobial peptides tailored to the user's research

Francisco Ramos-Martín, Thibault Annaval, Sébastien Buchoux, Catherine Sarazin, and Nicola D'Amelio

DOI: 10.26508/lsa.201900512

Corresponding author(s): Prof. Nicola D'Amelio (Université de Picardie Jules Verne)

Review timeline:

Submission Date:	2019-08-01
Editorial Decision:	2019-09-11
Revision Received:	2019-10-11
Editorial Decision:	2019-11-06
Revision Received:	2019-11-07
Accepted:	2019-11-08

Scientific Editor: Andrea Leibfried

Transaction Report:

No Peer Review Process File is available with this article, as the authors have chosen not to make the review process public in this case.

1st Editorial Decision

11 September 2019

September 11, 2019

Re: Life Science Alliance manuscript #LSA-2019-00512-T

Prof. Nicola D'Amelio
Université de Picardie Jules Verne
Bâtiment des poulies, 10, Rue Baudelocque
Gauche, rez-de-chaussée
Amiens, Hauts de France 80039
France

Dear Dr. D'Amelio,

Thank you for submitting your manuscript entitled "ADAPTABLE: a comprehensive web platform of antimicrobial peptides tailored to the user's research" to Life Science Alliance. The manuscript was assessed by expert reviewers, whose comments are appended to this letter.

As you will see, the reviewers think that your tool is valuable to the community, pending that current weaknesses get addressed in revision. We would thus like to invite you to submit a revised version of your work, addressing the individual concerns raised by the reviewers. Importantly, the main concern of reviewer #2 regarding the necessity to filter for only validated sequences needs to get addressed in a good way.

Thank you for this interesting contribution to Life Science Alliance. We are looking forward to receiving your revised manuscript.

Sincerely,

Andrea Leibfried, PhD
Executive Editor
Life Science Alliance

Meyerhofstr. 1
69117 Heidelberg, Germany
t +49 6221 8891 502
e a.leibfried@life-science-alliance.org
www.life-science-alliance.org

November 6, 2019

RE: Life Science Alliance Manuscript #LSA-2019-00512-TR

Prof. Nicola D'Amelio
Université de Picardie Jules Verne
Bâtiment des poulies, 10, Rue Baudelocque
Gauche, rez-de-chaussée
Amiens, Hauts de France 80039
France

Dear Dr. D'Amelio,

Thank you for submitting your revised manuscript entitled "ADAPTABLE: a comprehensive web platform of antimicrobial peptides tailored to the user's research". Please excuse the delay in getting back to you.

I appreciate the introduced changes and am happy to accept your manuscript in principle for publication here. Before sending you the official acceptance letter, please log into our submission system one more time to fill in the electronic license to publish form. Your manuscript number will change to LSA-2019-00512-TRR, please make sure to move all manuscript files to this new version.

A. FINAL FILES:

B. MANUSCRIPT ORGANIZATION AND FORMATTING:

Sincerely,

Andrea Leibfried, PhD
Executive Editor
Life Science Alliance
Meyershofstr. 1
69117 Heidelberg, Germany
t +49 6221 8891 502
e a.leibfried@life-science-alliance.org
www.life-science-alliance.org

November 8, 2019

RE: Life Science Alliance Manuscript #LSA-2019-00512-TRR

Prof. Nicola D'Amelio
Université de Picardie Jules Verne
Bâtiment des poulies, 10, Rue Baudelocque
Gauche, rez-de-chaussée
Amiens, Hauts de France 80039
France

Dear Dr. D'Amelio,

Thank you for submitting your Resource entitled "ADAPTABLE: a comprehensive web platform of antimicrobial peptides tailored to the user's research". It is a pleasure to let you know that your manuscript is now accepted for publication in Life Science Alliance. Congratulations on this interesting work.

DISTRIBUTION OF MATERIALS:

Again, congratulations on a very nice paper. I hope you found the review process to be constructive and are pleased with how the manuscript was handled editorially. We look forward to future exciting submissions from your lab.

Sincerely,

Andrea Leibfried, PhD
Executive Editor

Life Science Alliance
Meyershofstr. 1
69117 Heidelberg, Germany
t +49 6221 8891 502
e a.leibfried@life-science-alliance.org
www.life-science-alliance.org